# Fat-Soluble Vitamers: Parent-Child Concordance and Population Epidemiology in the Longitudinal Study of Australian Children

**DOI:** 10.3390/nu14234990

**Published:** 2022-11-24

**Authors:** Githal Randunu Porawakara Arachchige, Chris James Pook, Beatrix Jones, Margaret Coe, Richard Saffery, Melissa Wake, Eric Bruce Thorstensen, Justin Martin O’Sullivan

**Affiliations:** 1The Liggins Institute, The University of Auckland, Auckland 1023, New Zealand; 2Department of Statistics, Faculty of Science, The University of Auckland, Auckland 1010, New Zealand; 3The Murdoch Children’s Research Institute, Parkville, VIC 3052, Australia; 4Department of Paediatrics, University of Melbourne, Parkville, VIC 3010, Australia; 5The Maurice Wilkins Centre, The University of Auckland, Auckland 1010, New Zealand; 6MRC Lifecourse Epidemiology Unit, University of Southampton, University Road, Southampton SO17 1BJ, UK

**Keywords:** Longitudinal Study of Australian Children, parent, fat-soluble vitamers, liquid chromatography, tandem mass spectrometry

## Abstract

Fat-soluble vitamers (FSV) are a class of diverse organic substances important in a wide range of biological processes, including immune function, vision, bone health, and coagulation. Profiling FSV in parents and children enables insights into gene-environment contributions to their circulating levels, but no studies have reported on the population epidemiology of FSV in these groups as of yet. In this study, we report distributions of FSV, their parent-child concordance and variation by key characteristics for 2490 children (aged 11–12 years) and adults (aged 28–71 years) in the Child Health CheckPoint of the Longitudinal Study of Australian Children. Ten A, D, E and K vitamers were quantified using a novel automated LC-MS/MS method. All three K vitamers (i.e., K1, MK-4, MK-7) and 1-α-25(OH)_2_D_3_ were below the instrument detection limit and were removed from the present analysis. We observed a strong vitamer-specific parent-child concordance for the six quantifiable A, D and E FSVs. FSV concentrations all varied by age, BMI, and sex. We provide the first cross-sectional population values for multiple FSV. Future studies could examine relative genetic vs. environmental determinants of FSV, how FSV values change longitudinally, and how they contribute to future health and disease.

## 1. Introduction

Vitamers are one of several related compounds that exhibit similar biological activity to a specific vitamin (e.g., retinol, retinoic acid and retinyl palmitate are vitamers of vitamin A). Fat-soluble vitamers (FSV) A, D, E, and K are micronutrients indispensable for growth, reproduction, and the sustenance of optimum health at all stages of life. 

The association of an individual’s characteristics, notably age, sex and BMI, with circulating concentrations of FSV has been assessed in a number of epidemiological studies (e.g., [1,2,3]). Generally speaking, bioavailability decreases with age mainly due to impaired digestion, malabsorption from the gastrointestinal tract, and chronic diseases (reviewed in [4,5]. This is complicated by diet, supplement intake (e.g., [6]), location and distribution of body fat deposits (e.g., [7]). Obese children and adolescents have low FSV concentrations (e.g., 25-hydroxyvitamin D and α-tocopherol) compared to individuals with healthy weight. This is attributed to complex interactions (e.g., between obesity and inflammation) that alter FSV deposition and absorption (e.g., [8,9,10]). 

Population-level dietary intake data are required to formulate appropriate public health and food safety recommendations for effective supplementation or fortification policies. Most current epidemiologic studies focus on low cost, easy to administer food frequency questionnaires (FFQs) to semi-quantitatively assess FSV intake (e.g., vitamin D intake in 296 healthy 6 to 14 Year old children [11]). However, FFQs are known to produce measurement errors and the effectiveness of correction methods, and the total amount of errors are poorly understood (reviewed in [12]). By contrast, direct quantitative analyses of FSV levels accurately identify and quantitate circulatory vitamer concentrations. Quantitative analyses of FSV have previously been performed using immunoassays due to their turnaround time, throughput, and ease of troubleshooting and operating (reviewed in [13]). We have previously developed a high throughput, robust LC-MS/MS method that is capable of quantifying eleven A, D, E and K vitamers with high specificity and sensitivity [14].

Growing up in Australia: The Longitudinal Study of Australian Children (LSAC) is Australia’s largest and only public representative children’s cohort study concentrating on cultural, economic, physical, and social effects on health, social, learning, and cognitive development. LSAC consists of a wide range of psychosocial and administrative data collected during the first decade of the study [15]. The context of the present study is primarily based on LSAC’s Child Health CheckPoint, composed of sophisticated health assessments and biological samples. These included anthropometry, fitness, time use, hearing, vision, respiratory, cardiovascular, and bone health. Biospecimens included blood, buccal swabs (also from second parent), saliva, urine, toenails, and hair. The population characteristics are described in detail in Clifford et al., 2019 [16]. Our objective was to quantify FSV concentrations in Australian children and parents. 

Current data on population-level circulatory concentrations of FSV are scarce and limited by the range of analyzed vitamers and sample size (see [17]). There has been no simultaneous analysis of all four FSV classes at the population level. In this descriptive paper, our objective was to (1) quantify and (2) report on the distributions of FSV in 11–12 year old children and their parents in the Child Health CheckPoint study of the Longitudinal Study of Australian Children (LSAC), and (3) analyse differences in distribution by generation, sex, age and BMI. 

## 2. Materials and Methods

### 2.1. Study Design, Ethical Approval and Consent 

The work undertaken in this article is based on the B cohort module (Child Health CheckPoint; Appendix A) nested between waves 6 and 7 of the Longitudinal Study of Australian Children (LSAC) [15,16,18], Australia’s largest and only nationally-representative children’s cohort study. Population characteristics of the LSAC’s CheckPoint cohort can be found in Appendix A. In brief, LSAC recruited a nationally-representative sample of 5107 (initial uptake 57%) Australian infants in 2004 using a two-stage random sampling design, and followed them up in biennial ‘waves’ of largely survey-based and administrative data collection up to 2015. 

Of 3513 families that agreed at LSAC wave 6 to be contacted for the Checkpoint study, 1874 (53%) families subsequently took part in the CheckPoint, nested between LSAC waves 6 and 7 of children aged 11–12 years, between February 2015 and March 2016. Children and one parent attended the CheckPoint at a specialised 3.5-h (in all of Australia’s major and large cities) or 2.5-h (smaller regional cities) assessment center. Both generations underwent sophisticated health assessments and biological samples, including anthropometry and venipuncture relevant to this article. The population characteristics are described in detail in Clifford et al., 2019 [16] and the data collection, processing and standard operating procedures are available on https://mcri.figshare.com/articles/journal_contribution/Longitudinal_Study_of_Australian_Children_s_Child_Health_CheckPoint_Data_User_Guide_December_2018/5687590/3 (accessed on 20 November 2022). The study was approved by the Australian Institute of family studies Ethics Committee and Royal Children’s Hospital (Melbourne, Australia) Human Research Committee (33225D). 

### 2.2. Anthropometry

Body composition and weight were measured using Bioelectrical Impedance Analysis (BIA) scales, which provide accurate results when measuring percent body fat and fat-free body mass [19] The Body Mass Index (BMI) was calculated using the formula (weight/(average height)^2^) from directly-assessed height and body weight. The age in years (= age in weeks/52) and sex was determined based on two questions; Q1: what gender is the child? Q2: Are you male or female? (https://mcri.figshare.com/articles/journal_contribution/Child_Health_CheckPoint_Rationale_document/7716587/3 (accessed on 20 November 2022).

### 2.3. Sample Collection

Venus blood samples (28 mL) were collected at the CheckPoint center from semi-fasted parents (mean (SD) fasting time of 4.4 (2.1) h) and children (fasting time 3.4 (2.4) h) using single venipuncture [16]. 

Blood was processed at the CheckPoint on-site laboratory into 0.5 mL aliquots. Up to six EDTA plasma, six lithium heparin plasma, and six serum aliquots were extracted per participant and processed on average one hour later (range 1 min to 3.8 h) before storage at −80 °C [16]. Samples were then batched transferred to permanent storage at the Murdoch Children’s Research Institute. A set of lithium heparin plasma samples (*n* = 2490) were shipped to the Liggins Institute, the University of Auckland, on dry ice, in thermosafe boxes, where samples were stored (−80 °C) until further use.

### 2.4. Sample Randomization

Prior to the LC-MS/MS analysis, heparin plasma samples were randomised on dry ice into 34 different assay batches with 74 samples per batch, according to Andraos et al. 2020 [20]. During the randomization, the parent-child pairs (1221 pairs) were kept together in the same assay, with randomised samples stored at 80 °C until use. 

### 2.5. Stock Preparation 

Stock solutions of all-trans-retinol, retinoic acid, 25-OH-D_3_, 1-α-25(OH)_2_-D_3_, α-tocopherol, γ-tocopherol, α-tocotrienol, K1, MK-4, MK-7_,_ and isotopically-labelled internal standards 100 ng/mL of ([^2^H]_8_ retinol, [^2^H]_7_ 25-OH-D_3_, [^13^C]_6_ α-tocopherol and [^13^C]_6_ phylloquinone were prepared in ethanol (4 °C on ice). Calibration standards containing 10 FSV (concentrations ranged from 0.32–14,000 ng/mL (Appendix A)) were prepared as a mixture in phosphate-buffered saline (PBS) containing 4% (*w*/*v*) bovine serum albumin (BSA). The calibration mixture was serially diluted (2-fold) to achieve the required concentrations. Acetonitrile was used as a solvent blank. Each heparin plasma sample was spiked with the internal standards at physiologically relevant concentrations. Three quality controls (QCs) were prepared using pooled lithium heparin plasma. The QCs were spiked with 10 FSV external standards at three concentrations that span the physiological range (Appendix A) [14]. 

### 2.6. FSV Extraction and Sample Preparation by Liquid Handling Robot

Double hexane extraction of the FSVs was performed between January 2021 and April 2021, as described [14], using an Eppendorf EpMotion liquid handling robot, with a built-in vacuum manifold (EpMotion 5075vt, Hamburg, Germany) and thermal mixer. Plasma FSV extraction was programmed using epBlue Client version 40.6.2.6 software. The protocol was optimized to extract the standard mix, plasma samples, blank, and QCs. The robot deck consisted of serially diluted calibration standards in ethanol (for the seven-point calibration curve), three different QC mixes, acetonitrile mixed with the internal standards, hexane, and isopropanol. Plasma samples were loaded on the EpMotion immediately after thawing in a 20 °C water bath (Appendix A).

### 2.7. LC-MS/MS Analysis of FSV 

FSV concentrations were measured using a Q-Exactive™ mass spectrometer (Thermo Fisher Scientific, Dreieich, Germany) equipped with a heated electrospray ionization source. The LC-MS/MS analysis and validation criteria have been described (Appendix A) [14]. In brief, plates containing the extracted FSVs were sealed and loaded into a UPLC system consisting of a PAL autosampler with refrigerated sample trays (CTC Analytics, Thermo), an Accela 1250 UHPLC pump (Thermo Fisher Scientific, Austin, Texas, USA) and a HotDog 5090 column oven (Thermo). Samples were introduced into a Kinetex C18 100 Å (100 × 2.1 mm) 1.7 µm analytical column (Phenomenex; Auckland, New Zealand) fitted with a KrudKatcher Ultra HPLC in-line filter (Phenomenex) to chromatographically separate the compounds. A flow of 300 µL/min starting at 5% methanol and 95% MilliQ^®^, both with 5mM ammonium formate and 0.1% formic acid. Analytes were eluted using a gradient to 90% methanol (1.5 min), followed by an isopropanol and methanol gradient (13 min). The sample injection volume was 15 µL and run time was 19 min. All quality controls had acceptable recovery and reproducibility according to the EMA guidelines [21].

### 2.8. Statistical Analyses

All statistical analyses were performed using R Version 3.6.1 (https://www.r-project.org/ (accessed on 10 October 2022)). FSV α-tocopherol in plates 1 and 7 fell outside the 2 Standard deviation range; therefore, these plates were removed from the study to minimise technical plate effects. Quality Control data used for the present analysis can be accessed through https://auckland.figshare.com/articles/dataset/Child_CheckPoint_the_LSAC_cohort_study_quality_control_QC_data_of_FSV_analysis_using_LC-MS_MS/19333520/2 accessed on 20 November 2022). The reported FSVs included retinol, retinoic acid, 25-OH-D3, 1-α-25(OH)_2_-D_3_, α-tocopherol, γ-tocopherol, α-tocotrienol, K1, MK-4, and MK-7. The plasma concentrations of one of the two D vitamers (1-α-25(OH)_2_-D_3)_ and three K vitamers (K1, MK-4, and MK-7) were below the instrument detection limit for most samples and were excluded from subsequent analyses. Therefore, this analysis of FSV includes no K vitamers. 

The *lme4* package [22] in R was used to create mixed models to test the effects of generation (parent vs. child), sex (male vs. female), and family (shared gene-environment). Likelihood ratio tests were used to assess the importance of fixed effects for generation, sex, and their interaction in the presence of the family random effect. The familial concordance within parent-child dyads was also tested using Pearson’s correlations. 

Generalized additive models (GAM) with integrated smoothness estimation were used to calculate age and BMI effects for each variable in parents and children separately. Thin penalized regression splines were used to model the relationship, with smoothing parameters selected by generalized cross validation, as implemented in the mgcv package [23]. Due to the narrow age distribution in children (11–12 years), we only tested for age-specific differences in parents (28–71 years). BMI effects were tested for children and parents. Subjects with FSV concentrations beyond two standard deviations (SD) of the age- and BMI-specific means were considered as outliers and removed from the study. The R scripts used for this analysis can be accessed through https://auckland.figshare.com/articles/software/R_Scripts_Characterising_Fat-Soluble_Vitamer_profiles_in_Child_CheckPoint_of_the_LSAC_cohort_study/19333607/4 (accessed on 20 November 2022).

## 3. Results

Appendix A shows the flow of participants though LSAC and CheckPoint to the final analytic participant samples. FSVs were measured in plasma samples collected from children (11–12 years) and their parents as part of the LSAC’s CheckPoint study between February 2015 and March 2016 [16,24]. Concentrations of 6 FSV (retinol, 25-OH-D3, α-tocopherol, γ-tocopherol, and α-tocotrienol) were detectable and able to be measured for 2490 participants, comprising 1166 children (49% male), and 1324 parents (87% females, predominantly the biological mothers of children) (Table 1). Appendix A shows the distributions of parent and child BMI and of parent age.

### 3.1. FSV Profiles Are Concordant between Children and Their Parents 

Concentrations for retinoic acid, 25-OH-D_3,_ α-tocopherol, γ-tocopherol, α-tocotrienol and retinol show moderate to strong concordance (0.18 ≤ R ≤ 0.57; Pearson’s correlation) between children and parents from the same family (Figure 1).

### 3.2. FSV Profiles from the Same Family Are Generation and Sex-Dependent

Sex-and generation-specific effects were identified for the FSVs (Figure 2 and Appendix A). 25-OH-D_3_ concentrations were significantly higher (*p* < 0.01) in males (child-male and parent-male) than females (child-female and parent-female). The opposite was observed for γ-tocopherol (*p* = 0.02). There was no observed sex effect for α-tocotrienol (*p* = 0.4).

Compared to parents, the children had significantly higher 25-OH-D_3_ (*p* < 0.001) and α-tocotrienol (*p* = 0.05) plasma concentrations. By contrast, the γ-tocopherol concentration (*p* < 0.001) was higher in parents than children. Notably, the concentrations of retinol, retinoic acid, and α-tocopherol were significantly higher in both male and female parents than in their children (Figure 2 and Appendix A).

In the parent population, men had higher retinol (*p* < 0.001) and retinoic acid (*p* < 0.001) levels, while women had higher α-tocopherol (*p* < 0.001) ones. There was no observed sex effect in the child population for retinol, retinoic acid, or α-tocopherol (Figure 2 and Appendix A).

### 3.3. FSV Concentrations Are Age-Dependent

Higher retinol (*p* < 0.001), retinoic acid (*p* < 0.01) and α-tocopherol (*p* = 0.01) concentrations were seen in older parents (Figure 3). The effect for retinol and retinoic acid was non-linear, with a relatively flat trend until around age 45, before a rapid increase (Figure 3). By contrast, there were no significant associations between age and 25-OH-D_3_, γ-tocopherol or α-tocotrienol plasma concentrations in parents (Figure 3).

### 3.4. FSV Concentrations Are BMI Dependent

Generalized additive models identified associations between the majority of FSV plasma concentrations and BMI in parents and children (Figure 4). Parents with higher BMI had higher plasma concentrations of retinoic acid (*p* < 0.01), α-tocotrienol (*p* < 0.01) and γ-tocopherol (*p* < 0.001), but lower retinol (*p* = 0.03) and 25-OH-D_3_ (*p* < 0.001). In parents, the decline in retinol and increase in α-tocotrienol begin around BMI = 30, with relatively flat trends at lower BMIs (Figure 4). Parental α-tocopherol plasma concentrations were not significantly associated with BMI (Figure 4).

Similar to what was observed in parents, children with higher BMI had higher plasma concentrations of retinoic acid (*p* < 0.01) and γ-tocopherol (*p* < 0.001) and lower 25-OH-D_3_ (*p* < 0.001). In both children and adults, plasma retinol concentrations (*p* < 0.001) rose with BMI to around a BMI of 25–30; the falling retinol with greater BMI increments seen in parents was not evident in children, who did not include individuals with these very high BMIs. Plasma concentrations of α-tocopherol and α-tocotrienol were not associated with BMI in children (Figure 4).

## 4. Discussion

We used a novel multi-vitamer analytic platform to report for the first time on the population distributions of multiple fat-soluble vitamers simultaneously (A, D and E) in a large sample of 11–12 year old children and their parents. We identified sex, parent-child relatedness, age, and BMI as factors by which circulating plasma FSV levels varied within the LSAC CheckPoint cohort. To our knowledge, this is the first report of an interaction between sex, generation, and plasma FSV concentrations. Strong parent-child concordance was observed for plasma concentrations of all measured FSVs, supporting a shared gene-environment effect. Notably, the associations with age and BMI were not linear, with clear evidence for alterations in the relationship at particular ages and BMIs.

### 4.1. Strengths and Limitations

The strengths of this study included the very large, broadly population-representative sample of Australian adults and children, and the multiple FSVs paired in method and timing of collection, storage and bioassays and with the full range of accurately-measured BMI. Thus, we are confident in the age, sex, BMI and parent-child associations reported and the novelty of our findings for multiple FSV. Previously-published data are scant such that there are no population values for semi-fasted plasma samples from healthy children and adults for most of these vitamers. However, the plasma concentrations for the six quantified FSVs were within the previously reported endogenous levels of these vitamers, allowing for the differences in study populations/cohorts (Appendix A: retinol [25,26]^,^, retinoic acid [27,28], 25-OH-D_3_ [29,30,31]_,_ α-tocopherol [31,32,33], γ-tocopherol [31,34], and α-tocotrienol [35,36,37]).

The LSAC study design resulted in several limitations in our analysis. Firstly, we were limited to a single semi-fasted plasma sample from each LSAC participant. Secondly, only 10% of attending parents were fathers, limiting our power to detect age and sex-specific differences for men. Thirdly, as the child participants were all aged between 11–12 years, we could not study age-associated variation in FSV concentrations in children. The socio-economic status of the LSAC cohort is slightly higher and more homogeneous than that seen across the general Australian population. Therefore, while almost all levels of advantage/disadvantage were included, our results may not generalize to those living at the greatest disadvantage.

Lastly, FSVs are susceptible to degradation during extended storage at −70 to −80 °C [38,39,40,41,42], which probably explains why most measurements for several vitamers were below the instrument detection limit (i.e., 1-α-25(OH)_2_-D_3_ was measured in 5.9% of the samples; K1 in 8.8%; MK-4 in 3%; and MK-7 in 1.6%) and were thus excluded from our analysis. Furthermore, the mean values for the three E vitamers were at the lower end of values previously reported in smaller stand-alone studies (see Appendix A); it is not clear whether this also represents some degradation, but even if so, this should not affect the findings regarding parent-child concordance or variations by age, sex, or BMI. For the CheckPoint, the FSV bioassays on samples collected under field conditions in 2015 were not undertaken until 2020. For future studies, we recommend rigorous attention to the ideal storage of biosamples from the moment of collection (e.g., minimal light exposure) and the earliest possible conduct of bioassays to avoid the degradation due to processing, short and long-term storage conditions.

### 4.2. Interpretation

Parent-child dyads showed strong and consistent associations for all six A, D and E vitamers, with R^2^ ranging from 0.18 to 0.57 and tight confidence intervals. This suggests that plasma FSV levels are both heritable and environmental (e.g., parent-child similarities in dietary, activity and sunlight exposures), though we cannot tease these influences apart here.

We observed specific associations between plasma FSV concentrations and participants’ age and sex. The lack of a generally observable pattern for these associations indicates a complex interaction between the specific FSV and developmental and environmental factors (e.g., supplementation, sun exposure, sunscreen use, seasonal effect, and physical exercise). For example, absolute plasma concentrations and the ratio of α and γ-tocopherol have been previously shown to decline with age [43]. In contrast, we observed that plasma α-tocopherol, but not γ-tocopherol, levels increased with age in Australian parents. However, the relationship between age and γ-tocopherol was non-linear and may have been affected by outliers. We also observed a gradual increase in plasma retinol concentrations with age—although the rate increases at about 45 years of age. Both α tocopherol and retinol concentrations have been shown to be directly correlated with ‘healthy’ nutritional choices (e.g., diet, fiber, intrinsic sugars, potassium and vitamin supplements and inversely with sucrose) (e.g., [43,44,45]). Ideally, our study would be replicated in other large population-based studies and explore hypotheses regarding how FSV in different population sectors vary by dietary intake and physical activity (beyond the scope of this descriptive study).

Associations of FSV levels with BMI were largely (retinoic acid, 25-OH-D_3_, α-tocopherol, γ-tocopherol, α-tocotrienol) consistent in parents and children, but varied in directionality (positive, null and negative) for the different FSVs. Again, the associations were largely non-linear with evidence for inflection points that were specific to each FSV and may reflect changes in sample numbers. The association between adipose levels and vitamin D concentrations has been observed previously [46,47]. However, the associations between the other FSV and body fat have not been well characterised. We contend that the non-linear inter-relationships between these FSVs and BMI should be investigated using cohorts that span the full range of BMI values within a limited age range.

## 5. Conclusions

We have simultaneously measured and identified significant parent-child concordance for plasma FSV concentrations within individuals from the CheckPoint study of the Longitudinal Study of Australian Children (LSAC). Our results highlight that FSV concentrations are concordant in parent-child dyads, but that they vary by sex, BMI, and adult age with size and the non-linear nature of the associations tending to differ by vitamer. As well as the replication of these descriptive findings, future work in this and other cohorts could examine causal hypotheses related to dietary patterns (ideally with very accurate dietary intake data, notwithstanding the challenges of measurement), physical activity and body composition, including segmental adiposity.

## Figures and Tables

**Figure 1 nutrients-14-04990-f001:**
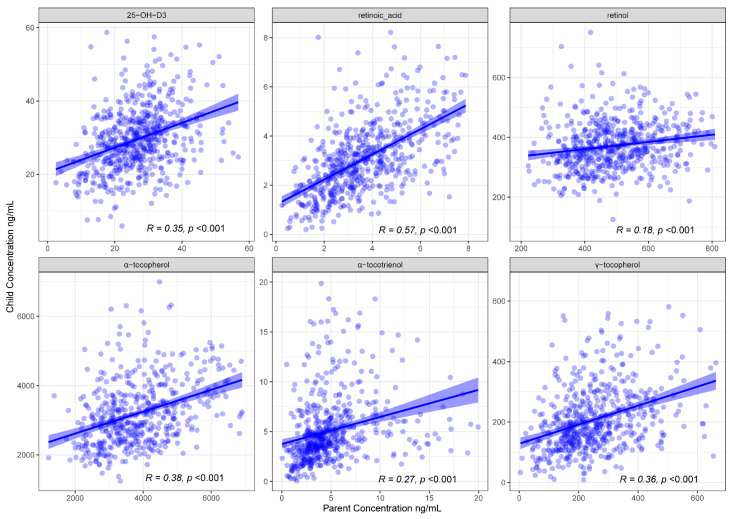
Scatter plots of parent-child FSV concentrations showing parent-child concordance. Linear models were fitted (R and *p* values as shown) with a 95% confidence interval.

**Figure 2 nutrients-14-04990-f002:**
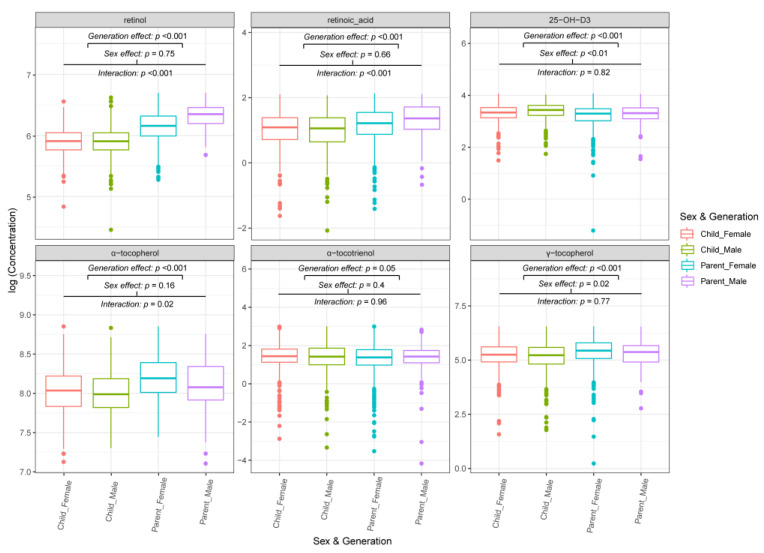
Box plots showing FSV concentrations by sex and generation. Outliers that were >2 standard deviations from the mean have been excluded from statistical analysis and the box plots. Mixed models were used to calculate *p*-values. The significance of the sex and interaction effect indicates whether the effect of sex differed between parents and children. A significant sex effect without significant interaction indicates that the sex effect is similar in the two generations. Where the interaction was significant, the sex effect was seen in adults only, without significant differences between male and female children.

**Figure 3 nutrients-14-04990-f003:**
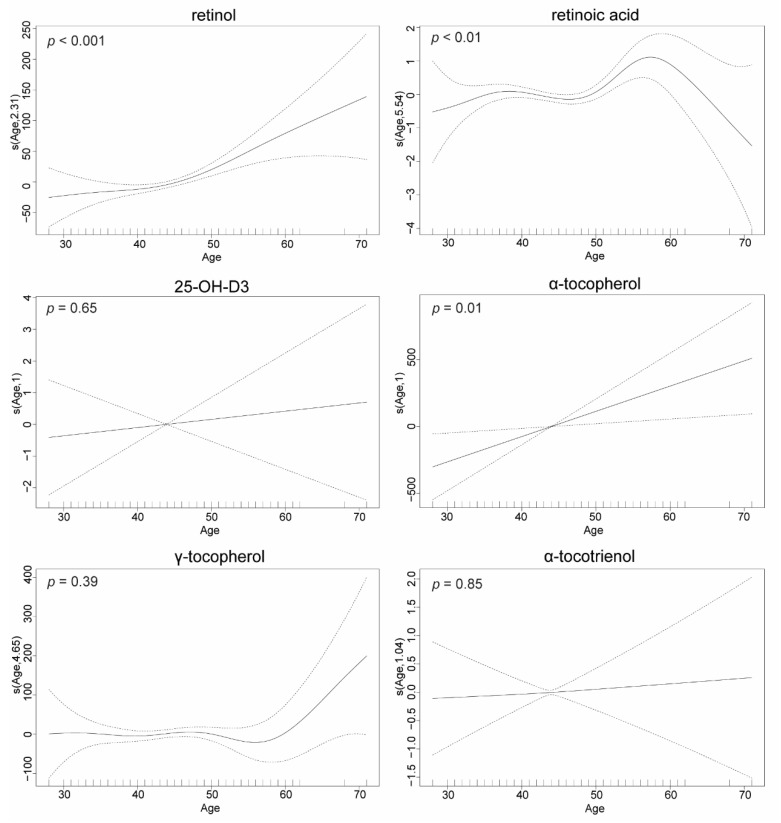
Generalized additive models of age effect on FSV concentrations. The number in s(Age, number) is the degree of freedom selected by the generalised cross-validation. When the number is one the procedure has selected a linear fit for the model. The dotted lines are estimate +/− 2 standard errors. The smooth functions are constrained to sum to zero over the observed covariate values; when a linear model is selected, this means that the function must be zero at the mean of the observed covariate values (0 uncertainty). The extreme ends of the curves are influenced by a small number of data points with unusual covariate values; this is reflected in the large window of uncertainty for these regions.

**Figure 4 nutrients-14-04990-f004:**
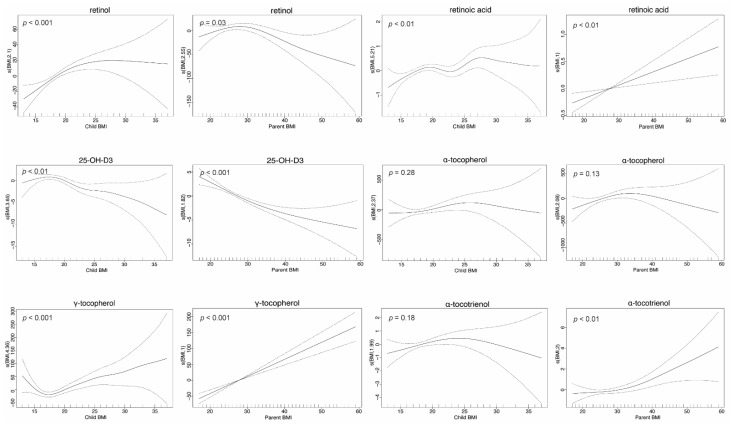
Generalized additive models of the BMI effect on FSV concentrations. The number in s (BMI, number) is the degrees of freedom selected by the generalised cross-validation. When the number is one the procedure has selected a linear fit for the model. The dotted lines are estimate +/− two standard errors. The smooth functions are constrained to sum to zero over the observed covariate values; when a linear model is selected, this means the function must be zero at the mean of the observed covariate values (0 uncertainty). The extreme ends of the curves are influenced by a small number of data points with unusual covariate values; this is reflected in the large window of uncertainty for these regions.

**Table 1 nutrients-14-04990-t001:** FSV concentrations within the CheckPoint cohort differed by sex and generation. n, number of samples; Gen, generation; Sex, biological sex.

Vitamin	Vitamer	Sex	n	Mean Concentration (ng/mL)	SD (ng/mL)	Gen	n	Mean Concentration (ng/mL)	SD (ng/mL)	Sex & Gen	n	Mean Concentration (ng/mL)	SD (ng/mL)
*A*	retinol	Female	1733	452	128	Child	1154	375	84.4	female child	596	375	85.3
										female parent	1137	492	129
		Male	731	425	133	Parent	1310	505	133	male child	558	374	83.5
										male parent	173	589	131
*D*	retinoic acid	Female	1722	3.53	1.72	Child	1145	3.14	1.57	female child	589	3.18	1.57
										female parent	1133	3.71	1.78
		Male	729	3.38	1.77	Parent	1306	3.78	1.83	male child	556	3.10	1.57
										male parent	173	4.28	2.07
	25-OH-D3	Female	1735	28.0	10.3	Child	1155	30.3	9.49	female child	595	29.1	9.30
										female parent	1140	27.4	10.8
		Male	733	30.8	9.55	Parent	1313	27.5	10.6	male child	560	31.6	9.52
										male parent	173	28.1	9.16
*E*	α-tocopherol	Female	1548	3702	1275	Child	1017	3173	937	female child	527	3233	952
										female parent	1021	3945	1351
		Male	641	3222	1032	Parent	1172	3899	1346	male child	490	3109	916
										male parent	151	3590	1277
	α-tocotrienol	Female	1709	5.39	5.57	Child	1150	5.70	5.17	female child	590	5.64	5.17
										female parent	1119	5.26	5.77
		Male	731	5.57	4.80	Parent	1290	5.21	5.50	male child	560	5.77	5.17
										male parent	171	4.92	3.21
	γ-tocopherol	Female	1742	255	168	Child	1160	222	157	female child	599	229	174
										female parent	1143	269	164
		Male	733	221	146	Parent	1315	265	165	male child	561	215	136
										male parent	172	244	172

## Data Availability

Data described in the article will be made available upon request after application and approval by our teams.

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
