# Peer review of "Fat-Soluble Vitamers: Parent-Child Concordance and Population Epidemiology in the Longitudinal Study of Australian Children"

_nutrients, 2022, doi:10.3390/nu14234990_

Round 1

Reviewer 1 Report

Thank you for the opportunity to review the very interesting  manuscript  " Fat-Soluble Vitamers: Parent-child concordance and population epidemiology in the Longitudinal Study of Australian Children ”, by  G.R.P. Arachchige1, C.J.Pook1, Beatrix Jones, M.Coe1, R. Saffery, M. Wake1, E.B. Thorstensen and  J.M. O'Sullivan.

The authors reported very interesting data on distribution of fat soluble vitamers in 2490 children (aged 11-12 years) and adults (aged 28–71 years)  obtained on the basis of automated LC-MS/MS method. They were the first cross-sectional population values for multiple vitamers.

They  observed a strong vitamer-specific parent-child concordance for the six quantifiable A, D and E vitamers concentrations all varied by age, BMI, and sex. 

All parts of the paper are clear and informative. I have one doubt that every reader of the article will understand the difference between vitamin and vitamer. In my opinion is worth to add such an explanation at the beginning of the paper.

The conclusions are consistent with the results presented and they all address to the objectives of the research.

It is worth emphasizing that authors paid attention to  limitations of this study, among them the lack of the data on the family dietary patterns, quantitative food and food supplements intake which may influence the content of the tested compounds.

The  further research are highly advisable and expected.

Author Response

We thank referee 1 for their review and very positive comments about our manuscript, including the consistency between our conclusions and results and our awareness of the study limitations.

  • I have one doubt that every reader of the article will understand the difference between a vitamin and vitamer. In my opinion, it is worth to add such an explanation at the beginning of the paper.
  • We have included the following sentence at the beginning of the manuscript.
    • "Vitamers are one of several related compounds that exhibits similar biological activity to a specific vitamin (e.g., retinol, retinoic acid and retinyl palmitate are vitamers of vitamin A)"

Reviewer 2 Report

In my opinion the manuscript entitled „Fat-Soluble Vitamers: Parent-child concordance and population epidemiology in the Longitudinal Study of Australian Children” presents an interesting data, but the manuscript requires major corrections.

 My comments:

Introduction

v  The Introduction is too short and poorly written, requires a significant improvement please expand this Section.

v  The use of questionnaires (FFQs) enables very simple, quick and non-invasive obtaining information on vitamin intake, while blood sampling does not belong to such methods. What, according to the authors, are the advantages and disadvantages of the presented method?

v  Line 67-68 Authors wrote that: “Most current epidemiologic studies focus on low cost, easy to administer food frequency questionnaires (FFQs) to semi-qualitatively assess FSV intake (e.g. vitamin intake in children aged 4-5 years.”, but the present study involved children aged 11-12 years - please correct your sentence.

v  Please provide the aim of this study, not what was presented in the manuscript.

Materials and methods

v  In my opinion, this section is described in an incomprehensible way.

v  Please provide the inclusion/exclusion criteria for this study.

v  Please explain what is the Child Health CheckPoint study of the Longitudinal Study of Australian Children (LSAC)?

v  Please describe in detail the recruitment of patients for this study.

v  How was the sample size for the test calculated? Authors should complete this information.

v  I suggest you put a scheme of this study.

Results

v  Please provide the characteristics of the studied group (children, parents).

v  Tables were prepared contrary to the guidelines of the journal.

v  Table 3 - the results presented in this table are incomprehensible.

v  Table 4 - the results presented in this table are incomprehensible.

v  The description of the results requires correction - description should be below the table / figure.

v  What was the state of health of the patients (other conditions, e.g. hypertension, diabetes, etc.)?

v  What was the assessment of the patients' nutrition? There are no results on the analysis of the patients' diet.

v  How was the body weight status determined? Were anthropometric measurements carried out? Please complete this information.

v  Did Authors analyze the physical activity of the study participants?

Discussion

v  This section is also poorly written and needs some significant corrections.

v  Please refer your own results to the results of other authors.

v  Please indicate possible reasons for the observed dependencies.

References

ü  The way of citation needs to be improved.

Author Response

We thank Referee 2 for their comments. However, we believe that this referee has read this manuscript from the perspective that we undertook the Checkpoint and LSAC studies. Instead, this paper is part of a series of papers that are targeting specific questions within the prospective longitudinal study of Australian children. All the details of the LSAC and Checkpoint studies have been previously published and are open access. We have included the following paragraph in our introduction to clarify this.

"Growing up in Australia: The Longitudinal Study of Australian Children (LSAC) is the Australia’s largest and only public representative children’s cohort study con-centrating on cultural, economic, physical, and social impacts on health, social, learning, and cognitive development [26]. LSAC consists of a wide range of psycho-social and administrative data collected during the first decade of the study. The context of the present study is primarily based on LSAC’s Child Health CheckPoint which included , composed of sophisticated health assessments and biological sam-ples on children aged 11-12 years [27]. These assessments included anthropometry,  fitness, time use, hearring, vision, respiratory, cardiovascular, and bone health. Bio-specimens included blood, buccal swabs (also from second parent), saliva, urine, toe-nails and hair [28]. The population characteristics are described in detail in Clifford et al., 2019 [26]. Our objective was to quantify FSV concentrations in Australian chil-dren and their parents."

  • The Introduction is too short and poorly written, requires a significant improvement please expand this Section.
  • We have included a paragraph on the LSAC study – see above. We believe that this improves the introduction significantly. 
  • The use of questionnaires (FFQs) enables very simple, quick and non-invasive obtaining information on vitamin intake, while blood sampling does not belong to such methods. What, according to the authors, are the advantages and disadvantages of the presented method?
  • We have included the following statement in the manuscript to clarify our position on this.

"However, FFQs are known to produce measurement errors and, the effectiveness of correction methods and the total amount of error are poorly understood (reviewed in 23)."

  • Line 67-68 Authors wrote that: "Most current epidemiologic studies focus on low cost, easy to administer food frequency questionnaires (FFQs) to semi-qualitatively assess FSV intake (e.g. vitamin intake in children aged 4-5 years.", but the present study involved children aged 11-12 years - please correct your sentence.
  • We have corrected this error. The sentence now states

"Most current epidemiologic studies focus on low cost, easy to administer food frequency questionnaires (FFQs) to semi-qualitatively assess FSV intake (e.g. vitamin D intake in 296 healthy 6 to 14 Year old children 22."

  • Please provide the aim of this study, not what was presented in the manuscript.
  • We have changed the text to read:

"In this descriptive paper, our objective was to (1) quantify and report on the distributions of FSV in 11-12 year old children and their parents in the Child Health CheckPoint study of the Longitudinal Study of Australian Children (LSAC), and (2) analyse differences in distribution by generation, sex, age and BMI."

  • In my opinion, the methods section is described in an incomprehensible way.
  • We have followed standard conventions for describing the methods. We contend that this issue has arisen due to a misunderstanding of the purpose of the manuscript and an expectation to see a full description of methods employed across the longitudinal study that is associated with the samples in this manuscript. The papers describing the LSAC and Checkpoint study have been referenced throughout the manuscript.
  • Please provide the inclusion/exclusion criteria for this study.
  • These criteria are outlined in the papers that describe the Checkpoint study, as referenced in the methods section.
  • Clifford, S. A.; Davies, S.; Wake, M. Child Health CheckPoint: Cohort Summary and Methodology of a Physical Health and Biospecimen Module for the Longitudinal Study of Australian Children. BMJ Open 2019, 9 (Suppl 3), 3–22. https://doi.org/10.1136/bmjopen-2017-020261.
  • Edwards, B. Growing up in Australia: The Longitudinal Study of Australian Children. Fam. Matters 2013, 91 (1), 7–17. https://doi.org/10.3316/INFORMIT.030635160763357.
  • Andraos, S.; Jones, B.; Wall, C.; Thorstensen, E.; Kussmann, M.; Smith, D. C.; Lange, K.; Clifford, S.; Saffery, R.; Burgner, D.; Wake, M.; O'Sullivan, J. Plasma b Vitamers: Population Epidemiology and Parent-Child Concordance in Children and Adults. Nutrients 2021, 13 (3), 1–15. https://doi.org/10.3390/nu13030821.
  • Edwards, B. Growing up in Australia: The Longitudinal Study of Australian Children: Entering Adolescence and Becoming a Young Adult. Fam. Matters 2014, No. 95, 5–14. https://doi.org/10.3316/IELAPA.837375544483712.
  • Please explain what is the Child Health CheckPoint study of the Longitudinal Study of Australian Children (LSAC)?
  • We have expanded our explanation of LSAC from what was originally in the methods section by including a paragraph in the introduction.

" Growing up in Australia: The Longitudinal Study of Australian Children (LSAC) is the Australia’s largest and only public representative children’s cohort study con-centrating on cultural, economic, physical, and social impacts on health, social, learning, and cognitive development [26]. LSAC consists of a wide range of psycho-social and administrative data collected during the first decade of the study. The context of the present study is primarily based on LSAC’s Child Health CheckPoint which included , composed of sophisticated health assessments and biological sam-ples on children aged 11-12 years [27]. These assessments included anthropometry,  fitness, time use, hearring, vision, respiratory, cardiovascular, and bone health. Bio-specimens included blood, buccal swabs (also from second parent), saliva, urine, toe-nails and hair [28]. The population characteristics are described in detail in Clifford et al., 2019 [26]. Our objective was to quantify FSV concentrations in Australian chil-dren and their parents."

  • Please describe in detail the recruitment of patients for this study.
  • Recruitment into this study is described in the manuscripts that describe the LSAC and Checkpoint cohorts. We also included a cohort diagram in Figure S1.
    1. How was the sample size for the test calculated? Authors should complete this information.
  • The sample size is described in the papers that describe the original cohort.
    1. I suggest you put a scheme of this study.
  • We assume the reviewer is asking for a schema for LSAC. This, beyond the scope of our paper, is published and presented in the papers that are referenced from our manuscript.
    1. Please provide the characteristics of the studied group (children, parents).
  • Participant information was included in Figure S1. We have also included a histogram of the population structure – age, BMI and sex (Figure S2)
  • Tables were prepared contrary to the guidelines of the journal.
  • We are not sure what the referee is referring to exactly. We contend we have met the journal requirements.
  • Table 3 - the results presented in this table are incomprehensible.
  • We are unsure how to change this table given the fact that Referee 1 had no issue with the information.
  • Table 4 - the results presented in this table are incomprehensible.
  • We are unsure how to change this table given the fact that Referee 1 had no issue with the information in this table.
  • The description of the results requires correction - the description should be below the table/figure.
  • This is a type-setting editing issue. We followed the author's guidelines provided by the journal, and they do not specify the position of the description in the submitted manuscript.
  • What was the state of health of the patients (other conditions, e.g. hypertension, diabetes, etc.)
  • As stated in the methods "LSAC recruited a nationally-representative sample of 5107 (initial uptake 57%) Australian infants in 2004 using a two-stage random sampling design, and followed them up in biennial 'waves' of data collection up to 2015; 1874 families subsequently took part in the CheckPoint, nested between LSAC waves 6 and 7 at child age 11–12 years, between February 2015 and March 2016. Children and one parent attended the CheckPoint at a specialised 3.5-hour (in all of Australia's major and large cities) or 2.5-hour (smaller regional cities) assessment centre where both generations underwent a comprehensive suite of questionnaires, physical assessments and biosamples." Data and analyses on the basis of the health status is beyond the scope of this manuscript.
  • What was the assessment of the patients' nutrition? There are no results on the analysis of the patients' diet.
  • We acknowledge this shortcoming in the discussion where we stated

"This suggests that plasma FSV levels are both heritable and environmental (e.g., parent-child similarities in dietary, activity and sunlight exposures), though we cannot tease these influences apart here.

And

Ideally, our study would be replicated in other large population-based studies and explore hypotheses regarding how FSV in different population sectors vary by dietary intake and physical activity (beyond the scope of this descriptive study).

And

As well as replication of these descriptive findings, future work in this and other cohorts could examine causal hypotheses related to dietary patterns (ideally with very accurate dietary intake data, notwithstanding the challenges of measure-ment), physical activity and body composition including segmental adiposity."

  • How was the body weight status determined? Were anthropometric measurements carried out? Please complete this information.
  • We have included the following sentence into the method section to point the reader to the SOPs for all measurements.

"The Data collection, processing and standard operating procedures are available on http://www.checkpoint-lsac.mcri. edu.au)".

  • Did Authors analyze the physical activity of the study participants.
  • See response to Q14. We did not analyse the physical activity levels of the study participants in this study. This is beyond the scope of this study.
  • This section is also poorly written and needs some significant corrections.
  • We cannot answer this criticism as it lacks critical information to inform what and where we should make “significant corrections”.
  • Please refer your own results to the results of other authors.
  • We have put our results into context with other published studies throughout the discussion.
  • Please indicate possible reasons for the observed dependencies.
  • We have discussed and presented hypotheses for the observed dependencies in the discussion. As part of this we have suggested future studies to elucidate the connections.
  • The way of citation needs to be improved.
  • We are unsure to what the referee is referring. Citations are according to the journal requirements.

Reviewer 3 Report

This study aims to investigate the population epidemiology of FSV. In general, the topic is interesting and the data is unique. However, I am highly concerned about the statistical analyses. I think major revision is needed to conduct and present the results in a clear and convincing way. My specific comments are as follows:

1, The paper starts with an introduction of FSV deficiencies symptoms, which makes reader expect that this study aims to test the association between FSV deficiencies and health outcomes. However, this paper only compares FSV concentrations among children and their parents with different sex, age, and BMI. I suggest authors revise the introduction from the beginning to avoid misunderstanding.

2, FSV concentration is only calculated based on a one-time blood collection during a one-year period. How to guarantee that participants’ blood sample will not be affected by the time of venipuncture and the health/nutritious status of participants?

3, Table 1 and figure 2 present the same information in different ways. The pairwise comparison can also be added in Table 1. I suggest move one of them to the supplementary file.

4, The way of presenting mixed model results in Table 2 is very strange. I strongly suggest authors adopted a multivariable regression to detect the association between FSV concentrations and participants’ characteristics such as sex and generation.

5, The age effect on FSV concentrations in parents shown in Table 3 is not correct. Because the association between these two variables may be nonlinear, thus insignificant coefficients do not mean that they are not correlated. A better way to show how FSV concentration changes over age is using a nonparametric method such as the local polynomial regression.

6, The association between BMI and FSV concentration (Table 4) can also be better captured by a nonparametric method.

7, The association between FSV concentration and participants’ characteristics should be discussed in depth, since this is the main contribution of your study.

Author Response

This study aims to investigate the population epidemiology of FSV. In general, the topic is interesting and the data is unique. However, I am highly concerned about the statistical analyses. I think major revision is needed to conduct and present the results in a clear and convincing way. My specific comments are as follows:

  • We thank the referee for their comments. We have revised the statistical analyses in line with the referee’s comments. This has strengthened the paper.

1, The paper starts with an introduction of FSV deficiencies symptoms, which makes reader expect that this study aims to test the association between FSV deficiencies and health outcomes. However, this paper only compares FSV concentrations among children and their parents with different sex, age, and BMI. I suggest authors revise the introduction from the beginning to avoid misunderstanding.

  • We have removed the section on FSV deficiencies from the introduction to remove the confusion.

2, FSV concentration is only calculated based on a one-time blood collection during a one-year period. How to guarantee that participants’ blood sample will not be affected by the time of venipuncture and the health/nutritious status of participants?

  • We noted the following in our discussion of the limitations of the study. “The LSAC study design resulted in several limitations in our analysis. Firstly, we were limited to a single semi-fasted plasma sample from each LSAC participant.“ As with all studies of this nature, we are unable to guarantee that there is not an affect of time and or health/nutrient status of the participants. However, the LSAC study is “broadly population-representative sample of Australian adults and children “ and includes representatives of “almost all levels of advantage/disadvantage”.

3, Table 1 and figure 2 present the same information in different ways. The pairwise comparison can also be added in Table 1. I suggest move one of them to the supplementary file.

  • We have done as requested

4, The way of presenting mixed model results in Table 2 is very strange. I strongly suggest authors adopted a multivariable regression to detect the association between FSV concentrations and participants’ characteristics such as sex and generation.

  • Table 2 has been modified and moved to the Supplementary materials.

5, The age effect on FSV concentrations in parents shown in Table 3 is not correct. Because the association between these two variables may be nonlinear, thus insignificant coefficients do not mean that they are not correlated. A better way to show how FSV concentration changes over age is using a nonparametric method such as the local polynomial regression.

  • We have done as requested and have now introduced non-linear models to describe the data in figures 3 and 4.

6, The association between BMI and FSV concentration (Table 4) can also be better captured by a nonparametric method.

  • We have done as requested and have now introduced non-linear models to describe the data in figures 3 and 4.

7, The association between FSV concentration and participants’ characteristics should be discussed in depth, since this is the main contribution of your study.

  • This paper’s aims are descriptive, ie to report the distributions of multiple FSV within a single population sample. An important element of description is to report how distributions vary by common population characteristics such as age, sex and BMI, and whether children’s values correlate with those of their parents.  As such, we did not approach our analyses within a causal framework and believe that it would be premature to go beyond the Discussion as it stands.

Round 2

Reviewer 2 Report

Unfortunately Authors did not satisfactorily correct the manuscript.

My comments:

v  The editorial preparation of the manuscript is incorrect. Authors did not correct the references, tables, supplementary file etc.

v  The introduction was not corrected. Authors included a fragment of the methodology in this part, the aim of the study should be indicated together with the justification, and the research hypotheses should be provided.

v  In each study, also in one that is part of another, the methodology should be precisely described, even if it was described in another publication. Therefore, Authors must describe this part.

v  Authors wrote: „Most current epidemiologic studies focus on low cost, easy to administer food frequency questionnaires (FFQs) to semi-qualitatively assess FSV intake (e.g. vitamin D intake in 296 healthy 6 to 14 Year old children 22." Please complete the references, if Authors write about the research, they should mention more than one example.

v  My previous comment: Please provide the inclusion/exclusion criteria for this study. Authors did not do it. Such ignorance of Authors is incomprehensible to me, because each research presented in the articles has the inclusion / exclusion criteria listed, even if they are part of larger studies.

v  Authors wrote that: „We are unsure how to change this table given the fact that Referee 1 had no issue with the information.” I suggest Authors read the Nutrients guidelines or look at the articles published in this journal, maybe then Authors will be able to improve the presentation of the results.

v  Reviewer asks for a schema of this study, not a schema for the LSAC.

v  Authors wrote that: „This is a type-setting editing issue. We followed the author's guidelines provided by the journal, and they do not specify the position of the description in the submitted manuscript.” I do not agree with this explanation because Authors did not prepare the tables correctly, hence the editorial problems.

v  My previous comment: How was the body weight status determined? Were anthropometric measurements carried out? Please complete this information. Authors did not do it. I regret that most of the information about this study the reader must find either in other studies or in a link to the website.

v  In addition, Authors' responses to my remarks were laconic and evasive for example: „Please refer your own results to the results of other authors.””We have put our results into context with other published studies throughout the discussion.”What exactly has been supplemented?

v  Authors wrote that: „We cannot answer this criticism as it lacks critical information to inform what and where we should make “significant corrections”. I suggest Authors re-read the guidelines for authors carefully. Alternatively, if Authors do not understand the structure of the discussion, you can also read sample research papers already published in this journal

v  Authors did not mark the changes made to the manuscript text, so it is not possible to check what was actually changed or corrected.

In my opinion, Authors do not understand the necessity to provide basic information about the conducted research, as well as a critical analysis of the obtained data and discussions.

Author Response

1) The editorial preparation of the manuscript is incorrect. Authors did not correct the references, tables, supplementary file etc.

> We have re-formatted the tables through-out the manuscript and in the supplementary material. We used the Style “Nutrients” within Mendeley for our referencing. We have checked this and it matches the format required by Nutrients.

2) The introduction was not corrected. Authors included a fragment of the methodology in this part, the aim of the study should be indicated together with the justification, and the research hypotheses should be provided.

>    In our first revision, we modified the introduction to include an introduction to Growing up in Australia The Longitudinal Study of Australian Children and the Child Health CheckPoint and thus provide important context for the reader.

>    This is a discovery study and is not hypothesis driven. We have changed the last paragraph of the introduction to read:

“ Current data on population-level circulatory concentrations of FSV are scarce and limited by the range of analysed vitamers and sample size (see [29]). There has been no simultaneous analysis of all four FSV classes at the population level. In this discovery study, we set out to (1) report on the distributions of FSV in 11-12 year old children and their parents in the Child Health Check-Point study of the Longitudinal Study of Australian Children (LSAC), and (2) analyse differences in distribution by generation, sex, age and BMI.”

3) In each study, also in one that is part of another, the methodology should be precisely described, even if it was described in another publication. Therefore, Authors must describe this part.

>    We agree with the referee, the methods section should include a clear and concise explanation of the procedures used to conduct the experiment. Such that, the experiment can be replicated. We contend that we have provided sufficient information and appropriate references to enable replication of these experiments. In addition, we had included the “Data Availability Statement: Data described in the article will be made available upon request after application and approval by our teams” in both the original and revised manuscripts. Thus, providing a mechanism through which the statistical analyses can be repeated using the full data set that was used in this manuscript.

4) Authors wrote: „Most current epidemiologic studies focus on low cost, easy to administer food frequency questionnaires (FFQs) to semi-qualitatively assess FSV intake (e.g. vitamin D intake in 296 healthy 6 to 14 Year old children 22." Please complete the references, if Authors write about the research, they should mention more than one example.

  • On further reflection this sentence was poorly written. We have altered the text to read “Food frequency questionnaires (FFQs) can be used as low cost, easy to administer, semi-qualitative methods for the assessment of FSV intake (e.g. vitamin D intake in 296 healthy 6 to 14 Year old children [22]”

5) My previous comment: Please provide the inclusion/exclusion criteria for this study. Authors did not do it. Such ignorance of Authors is incomprehensible to me, because each research presented in the articles has the inclusion / exclusion criteria listed, even if they are part of larger studies.

>    Section 2.1 and 2.2. describe the cohort and sample collection. The text states

The work undertaken in this article is based on the B cohort module (Child Health CheckPoint; Figure S1) nested between waves 6 and 7 of the Longitudinal Study of Australian Children (LSAC) [26,30]. Population characteristics of the LSAC’s CheckPoint cohort can be found in Figure S2. In brief, LSAC recruited a nationally-representative sample of 5107 (initial uptake 57%) Australian infants in 2004 using a two-stage random sampling design, and followed them up in biennial ‘waves’ of data collection up to 2015; In 2014-16, the Checkpoint study offered a special one-off physical health and biomarker module was offered to the LSAC children who were aged 11–12 year and their parents [31]; Of the 3513 families that agreed to be contacted for the Checkpoint study, 1874 parent–child dyads (53%) participated (Figure S1) [32,34].

1874 families subsequently took part in the CheckPoint, nested between LSAC waves 6 and 7 at child age 11–12 years, between February 2015 and March 2016. Children and one parent attended the CheckPoint at a specialised 3.5-hour (in all of Australia’s major and large cities) or 2.5-hour (smaller regional cities) assessment center where both generations underwent a comprehensive suite of questionnaires, physical assessments and biosamples [26,31,32].

6) Authors wrote that: „We are unsure how to change this table given the fact that Referee 1 had no issue with the information.” I suggest Authors read the Nutrients guidelines or look at the articles published in this journal, maybe then Authors will be able to improve the presentation of the results.

>    We have consulted the instructions to Authors and looked at Nutrients papers. We have modified the tables as requested.

7) Reviewer asks for a schema of this study, not a schema for the LSAC.

>    We have included a new figure that outlines the mass spectrometry method that was used in this manuscript (Figure S4). We consider that collectively Figures S3 and S4 outline the full methodology that was employed in the undertaking of this discovery study.

8) Authors wrote that: „This is a type-setting editing issue. We followed the author's guidelines provided by the journal, and they do not specify the position of the description in the submitted manuscript.” I do not agree with this explanation because Authors did not prepare the tables correctly, hence the editorial problems.

>    As stated in our response to point 6) above, we have consulted the instructions to Authors and looked at Nutrients papers. We have modified the tables as requested.

9) My previous comment: How was the body weight status determined? Were anthropometric measurements carried out? Please complete this information. Authors did not do it. I regret that most of the information about this study the reader must find either in other studies or in a link to the website.

>    We have included a new section that outlines all of the methods that were used for the anthropometric measures that were analysed as part of this study (methods Section 2.2). This text reads

“Body composition and weight were measured using Bioelectrical Impedance Analysis (BIA) scales, which provide accurate results when measuring percent body fat and fat-free body mass [33] The Body Mass Index (BMI) was calculated using the formula (weight/(average height)^2) from directly-assessed height and body weight. The age in years (= age in weeks/52) and sex was determined based on two questions; Q1: what gen-der is the child?, Q2: Are you male or female? (https://mcri.figshare.com/articles/journal_contribution/Child_Health_CheckPoint_Rationale_document/7716587)”

10) In addition, Authors' responses to my remarks were laconic and evasive for example: „Please refer your own results to the results of other authors. ”We have put our results into context with other published studies throughout the discussion.”What exactly has been supplemented?

>    We have not modified the discussion. We contend that the structure and content is appropriate for this study. We have followed suggested structures for writing discussions (1) that include a detailed exposition of the strengths and weaknesses of the study, and a critical assessment of our results in the context of existing studies. We have then completed this with a simple and clear conclusion that suggests future work.

      (1) Docherty M, Smith R. The case for structuring the discussion of scientific papers. BMJ. 1999 May 8;318(7193):1224-5. doi: 10.1136/bmj.318.7193.1224. PMID: 10231230; PMCID: PMC1115625.

11) Authors wrote that: „We cannot answer this criticism as it lacks critical information to inform what and where we should make “significant corrections”. I suggest Authors re-read the guidelines for authors carefully. Alternatively, if Authors do not understand the structure of the discussion, you can also read sample research papers already published in this journal

>    See answer to 10) above.

12) Authors did not mark the changes made to the manuscript text, so it is not possible to check what was actually changed or corrected.

>    We submitted a track changed version of the manuscript under the documents for review category.

13) In my opinion, Authors do not understand the necessity to provide basic information about the conducted research, as well as a critical analysis of the obtained data and discussions.

>          We decline to respond to this comment.

Reviewer 3 Report

Authors have addressed/answered most of my comments. I only have a few minor suggestions:

First, there are many errors in reference citation in section 3.2. Please fix these errors in the revision.

Second, some fitted lines in Figure 3 & 4 are very strange. I think more careful checking is needed.

Third, figure 3 & 4 indicate that the association between FSV concentrations and age/BMI changes at different age/BMI values. Thus the discussion should be more careful as the statistical significance also differs at different age/BMI values.

Author Response

First, there are many errors in reference citation in section 3.2. Please fix these errors in the revision.

  • We have checked all the references and have removed the links to the figures in the revision.

Second, some fitted lines in Figure 3 & 4 are very strange. I think more careful checking is needed.

  • We have checked the fitted lines in Figures 3 and 4. We have added the following text tot eh legends to explain those that look unusual.

“ The smooth functions are constrained to sum to zero over the observed covariate values; when a linear model is selected, this means the function must be zero at the mean of the observed covariate values (0 uncertainty). The extreme ends of the curves are influenced by a small number of data points with unusual covariate values; this is reflected in the large window of uncertainty for these regions.”

Third, figure 3 & 4 indicate that the association between FSV concentrations and age/BMI changes at different age/BMI values. Thus the discussion should be more careful as the statistical significance also differs at different age/BMI values.

  • We have modified the discussion to include the following sentences to address this.

“Notably, the associations with age and BMI were not linear, with clear evidence for alterations in the relationship at particular ages and BMIs.”

“However, the relationship between age and γ-tocopherol was non-linear and may have been affected by outliers. We also observed a gradual increase in plasma retinol concentrations with age – although the rate increases at about 45 years of age.”

“Again, the associations were largely non-linear with evidence for inflection points that were specific to each FSV and may reflect changes in samples numbers. The association between adipose levels and vitamin D concentrations has been observed previously [47,48]. However, the associations between the other FSV and body fat have not been well characterised. We contend that the non-linear inter-relationships between these FSVs and BMI should be investigated using cohorts that span the full range of BMI values within a limited age range.”
